

# Conservation and diversity in expression of candidate genes regulating socially-induced female-male sex change in wrasses

Jodi T. Thomas[1,2], Erica V. Todd[2], Simon Muncaster[3], P Mark Lokman[4], Erin L. Damsteegt[4], Hui Liu[2], Kiyoshi Soyano[5], Florence Gléonnec[2,6], Melissa S. Lamm[7], John R. Godwin[7] and Neil J. Gemmell[2]

[1] ARC Centre of Excellence for Coral Reef Studies, James Cook University, Townsville, Queensland, Australia
[2] Department of Anatomy, University of Otago, Dunedin, Otago, New Zealand
[3] Faculty of Primary Industries, Environment and Science, Toi Ohomai Institute of Technology, Tauranga, Bay of Plenty, New Zealand
[4] Department of Zoology, University of Otago, Dunedin, Otago, New Zealand
[5] Institute for East China Sea Research, Organization for Marine Science and Technology, Nagasaki University, Taira-machi, Nagasaki, Japan
[6] BIOSIT - Structure Fédérative de Recherche en Biologie-Santé de Rennes, Université Rennes I, Rennes, France
[7] Department of Biological Sciences and WM Keck Center for Behavioral Biology, North Carolina State University, Raleigh, NC, United States of America

Corresponding authors
Jodi T. Thomas,
jodi.thomas@my.jcu.edu.au
Erica V. Todd,
erica.v.todd@otago.ac.nz

## ABSTRACT

Fishes exhibit remarkably diverse, and plastic, patterns of sexual development, most striking of which is sequential hermaphroditism, where individuals readily reverse sex in adulthood. How this stunning example of phenotypic plasticity is controlled at a genetic level remains poorly understood. Several genes have been implicated in regulating sex change, yet the degree to which a conserved genetic machinery orchestrates this process has not yet been addressed. Using captive and in-the-field social manipulations to initiate sex change, combined with a comparative qPCR approach, we compared expression patterns of four candidate regulatory genes among three species of wrasses (Labridae)—a large and diverse teleost family where female-to-male sex change is pervasive, socially-cued, and likely ancestral. Expression in brain and gonadal tissues were compared among the iconic tropical bluehead wrasse (*Thalassoma bifasciatum*) and the temperate spotty (*Notolabrus celidotus*) and kyusen (*Parajulus poecilepterus*) wrasses. In all three species, gonadal sex change was preceded by downregulation of *cyp19a1a* (encoding gonadal aromatase that converts androgens to oestrogens) and accompanied by upregulation of *amh* (encoding anti-müllerian hormone that primarily regulates male germ cell development), and these genes may act concurrently to orchestrate ovary-testis transformation. In the brain, our data argue against a role for brain aromatase (*cyp19a1b*) in initiating behavioural sex change, as its expression trailed behavioural changes. However, we find that isotocin (*it*, that regulates teleost socio-sexual behaviours) expression correlated with dominant male-specific behaviours in the bluehead wrasse, suggesting *it* upregulation mediates the rapid behavioural sex change characteristic of blueheads and other tropical wrasses. However, *it* expression was not sex-biased in temperate spotty and kyusen wrasses, where sex change is more protracted and social groups may be less tightly-structured. Together, these findings

suggest that while key components of the molecular machinery controlling gonadal sex change are phylogenetically conserved among wrasses, neural pathways governing behavioural sex change may be more variable.

## INTRODUCTION

Most animals irreversibly differentiate as either male or female, yet some species exhibit remarkable sexual plasticity. This is true for teleost fishes, the only vertebrate lineage to display sequential hermaphroditism, in which individuals begin life as one sex but can change to the opposite sex sometime later in their life cycle (*Munday, Buston & Warner, 2006*; *Devlin & Nagahama, 2002*). Sex change is typically cued by changes in social structure or by reaching a threshold age or size (*Shapiro & Lubbock, 1980*; *Lee et al., 2001*), and characteristically involves radical changes in behaviour, external colouration and gonadal anatomy (*Warner & Swearer, 1991*; *Todd et al., 2016*). Three patterns are observed; protogyny (female-to-male), protandry (male-to-female), and bidirectional sex change (*Warner, 1984*). Protogyny is most common, although the widespread and patchy distribution of sequential hermaphroditism across the teleost phylogeny implies multiple evolutionary origins and frequent transitions to and from gonochorism (stable separate sexes) (*Mank, Promislow & Avise, 2006*).

Despite significant research effort, the genetic cascades that orchestrate sex change remain elusive (*Todd et al., 2016*). Numerous genes involved in vertebrate sexual development have been investigated for their potential roles in sex change (*Todd et al., 2016*). Genes that exhibit expression changes early on in sex change are of particular interest as proximal molecular regulators of the process. One such gene is *cyp19a1a*, encoding the aromatase enzyme that converts testosterone (T) to 17$\beta$-estradiol (E2) in the female gonad to maintain ovarian function (*Devlin & Nagahama, 2002*; *Tchoudakova & Callard, 1998*). Aromatase expression is rapidly arrested in transitioning females and this occurs in parallel with a sharp decline in plasma E2 levels and the onset of ovarian atresia (*Nakamura et al., 1989*). Treatment with aromatase inhibitors reliably induces complete sex reversal in teleosts, whereas co-administration with E2 is preventative (*Higa et al., 2003*; *Nozu, Kojima & Nakamura, 2009*; *Bhandari et al., 2005*; *Kroon & Liley, 2000*). Thus, arrested *cyp19a1a* expression may initiate gonadal sex change in protogynous species by interrupting a positive E2 feedback loop that in fishes maintains both feminising gene expression and ovarian function (*Todd et al., 2016*; *Guiguen et al., 2010*).

The most well-studied potential initiator of the male-specific expression pathway in sex-changing species is *dmrt1*, a gene that encodes a transcription factor critical for promoting male gonadal development in animals as diverse as flies and humans (*Herpin & Schartl, 2011a*). A paralogue of *dmrt1* (*dmy*) has also become the male sex-determining

gene in several fish species (*Matsuda et al., 2002*; *Nanda et al., 2002*; *Chen et al., 2014*). However, in protogynous hermaphrodites studied to date, changes in *dmrt1* expression regularly appear downstream of other genes, suggesting that *dmrt1* may be more important in progressing rather than initiating sex change (*Nozu et al., 2015*; *Todd et al., 2018*).

Anti-Müllerian hormone, Amh, a multifunctional member of the transforming growth factor-ß(TGF-ß) family, also plays a key role in regulating germ cell development in vertebrates, especially in males (*Josso, Di Clemente & Gouédard, 2001*; *Siegfried, 2010*; *Sekido & Lovell-Badge, 2013*). Amh is the male-determining factor in Patagonian pejerrey (*Odontesthes hatcheri*) (*Hattori et al., 2012*) and Nile tilapia (*Oreochromis niloticus*) (*Li et al., 2015*), while the Amh receptor, Amhr2, determines maleness in several species of *Takifugu* pufferfish (*Kamiya et al., 2012*). A transcriptome-wide expression analysis of bluehead wrasse found *amh* and *amhr2* to be the earliest male-pathway genes upregulated during female to male sex change, concurrent with arrested expression of *cyp19a1a* and prior to the appearance of male tissues (*Todd et al., 2018*). Expression of *amh* also increased during early protogynous sex change in ricefield eel (*Monopterus albus*) (*Hu et al., 2015*), and decreased during protandrous sex change in Red Sea clownfish (*Amphiprion bicinctus*) (*Casas et al., 2016*). Therefore, Amh is emerging as a key initiator of maleness in gonochoristic and sex-changing fish.

Most studies focus on gonadal gene expression, yet social cues for sex change are visual and induce rapid neurochemical changes in the brain to initiate behavioural responses that precede, and likely trigger, gonadal changes (*Larson, Norris & Summers, 2003*; *Semsar & Godwin, 2003*; *Godwin & Thompson, 2012*; *Lamm et al., 2015*). Teleost fishes are unique in having a duplicated, brain-specific paralogue of the aromatase gene, *cyp19a1b*, responsible for local oestrogen production that plays a key role in brain sexualisation (*Diotel et al., 2010*). Paralleling gonadal *cyp19a1a* activity, forebrain *cyp19a1b* expression is downregulated in transitioning female bluehead wrasse (*Todd et al., 2018*). Treatment with exogenous E2 also stimulates *cyp19a1b* expression and prevents behavioural sex change in this species (*Marsh-Hunkin et al., 2013*).

A further gene of growing interest is isotocin (*it*) (*Liu et al., 2016*; *Todd et al., 2017*). Homologous to mammalian oxytocin, *it* appears to regulate teleost sociosexual behaviours (*Goodson & Bass, 2000*; *Thompson & Walton, 2004*; *O'Connell, Matthews & Hofmann, 2012*; *Reddon et al., 2014*; *Reddon et al., 2012*; *Hellmann et al., 2015*; *Donaldson & Young, 2008*). Transcriptomic analyses in the bluehead wrasse have found forebrain *it* expression to be specific to terminal-phase males, implicating *it* in social dominance and sex change (*Liu et al., 2015*).

Protogyny is most pervasive, and likely ancestral, in labrid fishes (*Erisman et al., 2013*). The Labridae are the second largest marine fish family, encompassing the wrasses, parrotfish and hogfish with over 500 species in 70 genera (*Baliga & Law, 2016*; *Westneat & Alfaro, 2005*). Protogyny is best studied in wrasses, which present a powerful model to study the evolution and functioning of sex change and explore the degree to which molecular control of this process is conserved. Labrids have a characteristic lek-like mating system, and are often diandric with two colour morphs; initial phase (IP) individuals consist of similarly coloured females and less abundant primary males (female-mimics), which can

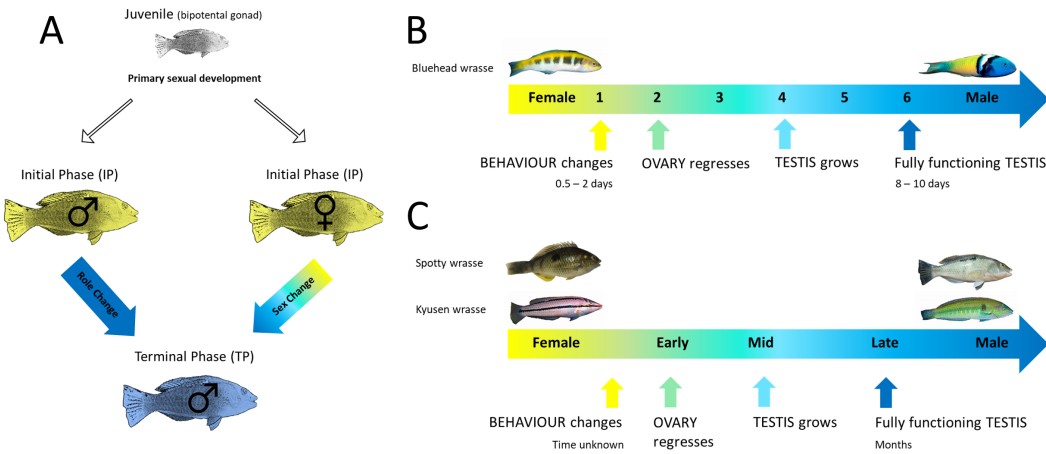

**Figure 1 Life history of protogynous wrasses.** (A) Generalised life cycle of protogynous fishes. Juveniles with a bipotential gonad undergo primary sexual development as either initial phase (IP) females or males. Terminal phase (TP) males develop via sex change by IP females, or role change by IP males, following appropriate social cues. Figure adapted from *Todd et al. (2017)*. (B) Progression of sex change in bluehead wrasse which is classified into 6 stages as previously described (*Nakamura et al., 1989*), and occurs remarkably fast; behavioural change occurs within 0.5–2 days and complete ovary-to-testis transformation is completed in 8–10 days (*Warner & Swearer, 1991*). (C) Progression of sex change in spotty and kyusen wrasses which is classified into early, mid and late stages, broadly corresponding to stages 2–3, 4 and 5–6 in the bluehead wrasse, respectively. Sex change in these seasonal breeders may take up to several months. Figure adapted from *Liu et al. (2016)*. Bluehead wrasse male image open access by Evan D'Alessandro, courtesy Oregon State University; bluehead wrasse female image open access; spotty male image by Jodi Thomas; spotty female image by Simon Muncaster; kyusen images with permission by Keoki Stender http://www.marinelifephotography.com.

sex or role change respectively, to replace the dominant terminal phase (TP) male upon its death or removal (Fig. 1A) (*Warner & Swearer, 1991*; *Kazancıoğlu & Alonzo, 2010*). Sex change occurs year-round in tropical wrasses, but follows a discreet spawning season in temperate species and occurs more slowly and from an already regressed ovary with low oestrogen production (*Muncaster, Norberg & Andersson, 2013*). Thus, an important question is whether aromatase downregulation plays a pivotal role initiating sex change in both tropical and temperate wrasses.

In this study, four candidate genes are evaluated as proximate regulators of protogynous sex change, in the gonad (*cyp19a1a*, *amh*) and brain (*cyp19a1b*, *it*), using a comparative approach across three diandric protogynous wrasses (Figs. 1B and 1C): the tropical Caribbean bluehead wrasse (*Thalassoma bifasciatum*), and the temperate New Zealand spotty wrasse (*Notolabrus celidotus*) and Japanese kyusen wrasse (*Parajulis poecilepterus*). We sought to (1) investigate whether evolutionarily conserved molecular mechanisms underlie protogynous sex change in wrasses, and (2) examine potential differences among tropical versus temperate species. Specifically, we were interested in the importance of changes in *aromatase* expression in initiating gonadal sex change and *isotocin* expression in initiating behavioural sex change in temperate species, in which sex change proceeds from post-spawning, already regressed ovaries, and in which social hierarchies may be less tightly structured.

## MATERIALS & METHODS

### Sample collection

*Experiment 1: social induction of sex change in wild bluehead wrasse*

Sex change was induced in wild bluehead wrasse social groups by social manipulation on patch reefs off the coast of Key Largo, Florida, between May and June 2014. These experiments are described in detail elsewhere (*Liu et al., 2015*). Three individuals representing each of six stages of sex change (described below), plus control females, IP males and TP males were used in the current study. Experiments were conducted in accordance with approval from North Carolina State University (12-069-0).

*Experiment 2: social induction of sex change in captive spotty wrasse*

A social manipulation experiment was conducted to induce sex change in captive female spotty wrasse between August and December 2016, towards the end of the spawning season and overlapping the period when sex change is documented in the wild (November–May) (*Jones, 1980*). Fish were collected from Tauranga Harbour (37°40′29′′S; 176°10′20′′E) by hook and line. Fifty IP fish were evenly distributed into groups across ten 500 L tanks containing recirculating seawater (35 ppt). IP individuals ranged from 149 mm–217 mm total length (TL) and were distributed such that each tank contained a hierarchy of different sized fish plus a single TP male (size range 222 mm–247 mm TL). Ambient light was available through semi-translucent roof panels and was supplemented with overhead fluorescent lighting (Sylvania Cool White De Luxe, Osram Sylvania Ltd) on a 12:12 light:dark daily cycle. Fish were fed a combination of thawed mussel, *Perna canaliculus*, and commercial marine fish feed (Ridley Aquafeed, Ridley Corporation) to satiation three times per week.

Following an acclimation period of three weeks, TP males were removed from five of the 10 tanks (day 0), creating a permissive social environment for sex change. As a control on day 0, the largest IP fish were removed from each of the control tanks, and also served as a baseline. Subsequently, the largest IP fish per tank was removed at day 30, 50, 60, 65, and 66 post TP male removal. Fish were immediately anaesthetised in an aerated bath containing 6 ml L$^{-1}$ 2-phenoxyethanol (Sigma Aldrich) before being euthanized by decapitation. A gonad section (mid-section of one paired gonad) and the whole brain were preserved in RNAlater (Invitrogen$^{TM}$, Thermo Fisher Scientific), chilled at $-18$ °C for 24 h, then stored at $-80$ °C until RNA extraction. An additional gonad section (mid-section of second gonad) was fixed in either Bouin's (testis and transitional gonads) or neutral-buffered formalin (ovary), and subsequently dehydrated by submersion in ethanol (70, 80, 96 and 100%) followed by xylene. Gonadal tissue was paraffin-embedded and 3 μm sections were stained with hematoxylin and eosin (H&E) for light microscopy (New Zealand Veterinary Pathology, Hamilton) to determine sexual status. Fish were collected with approval from the New Zealand Ministry of Primary Industries (593-3) and experiments were conducted in accordance with approval from the New Zealand National Animal Ethics Advisory (2015_02).

### Survey 1: opportunistic sampling of spotty wrasse

Seven fish were caught by hook and line off Portobello Wharf, Dunedin, New Zealand, and an additional seven fish were obtained from the nearby New Zealand Marine Studies Centre, during the non-breeding season between March and May 2013. Fish were euthanized with an overdose of benzocaine (0.3 g/L) and the brain and gonads dissected immediately. One gonad and the whole brain were preserved in RNAlater (Life Technologies, Inc.) on ice, before storage at −80 °C until RNA extraction. The second gonad was preserved in 10% formalin for histological analysis. Gonadal tissue was paraffin-embedded and 5 μm sections were stained with H&E to determine sexual status (Histology Services Unit, University of Otago). Experiments were conducted in accordance with approval from the New Zealand National Animal Ethics Advisory (92-10).

### Survey 2: wild-caught kyusen wrasse

Fish were caught by hook and line from Oomura Bay ($n = 2$) and Chijiwa Bay ($n = 29$), Kyushu Island, Japan, at the end of the breeding season between September and November 2010. Fish were euthanized with an overdose of 2-phenoxyethanol and the brain and gonads dissected out immediately. A gonad section and the brain were preserved in RNAlater (Life Technologies, Inc.) on ice, or flash frozen in liquid nitrogen, before storage at −80 °C until RNA extraction. An additional gonad section was preserved in Bouin's fixative for histological analysis. Gonadal tissue was paraffin-embedded and 5 μm sections were stained with H&E to determine sexual status. Experiments were conducted in accordance with approval from the Animal Care and Use Committee of the Institute for East China Sea Research, Nagasaki University, Japan (#15-06).

## Histological analysis of the gonad

For bluehead wrasse, transitioning fish were grouped into six stages as per the classification system of Nakamura et al. (1989). As seasonal breeders, female spotty and kyusen wrasses were classified as either non-breeding female (NBF) or breeding female (BF), depending on the presence of maturing oocytes. Transitioning animals were classified into early transitional (ET), mid transitional (MT) or late transitional (LT) stages (see Table S1). ET fish were distinguished from NBF by having elevated oocyte atresia, including degenerating previtellogenic oocytes, and typically contained nests of gonial germ cells (presumed to later become spermatogonia), masses of stromal cells and cellular debris (also present as yellow-brown bodies) as observed in other ET protogynous species (Muncaster, Norberg & Andersson, 2013; Bhandari et al., 2003) (Table S1). In spotty and kyusen wrasse, the ET, MT and LT stages broadly correspond to stages 2-3, 4 and 5-6 as outlined by Nakamura et al. (1989) and used to classify bluehead wrasse.

## RNA extraction

Due to samples being obtained from several sources and processed at different times, different extraction protocols were used and are summarised in Table S2. For spotty and bluehead brain samples, the hindbrain (corpus cerebelli, pons, and medulla) was removed prior to RNA extraction. The forebrain/midbrain was prioritised for analysis as it is

expected to contain key neural circuits involved in socially regulated sex change (*O'Connell & Hofmann, 2011*).

## Reverse transcription

Total RNA was quantified by Qubit 2.0 Fluorometer (Qubit RNA HS Assay Kit, Life Technologies), and RNA purity was measured by spectrophotometer (NanoDrop 2000c, ThermoFisher Scientic). Bluehead RNA was reverse transcribed in a Mastercycler Pro S thermal cycler (Eppendorf) with the following protocol: 37 °C (15 mins), 85 °C (5 s), 4 °C until removal. For spotty and kyusen, reverse transcription reactions were performed in a SureCycler 8800 (Agilent Technologies) with the following protocol: 25 °C (10 mins), 37 °C (120 min), 85 °C (5 mins), 4 °C until removal. Further details are provided in Table S2.

## Determination of gene sequences

Preliminary sequence data for four target genes (*cyp19a1a*, *amh*, *cyp19a1b*, and *it*) and three potential reference genes (*ef1a*, *18S*, and *g6pd*) were obtained from transcriptome assemblies for bluehead wrasse (*Liu et al., 2015*) and spotty wrasse (EV Todd & NJ Gemmell, 2015, unpublished data) representing gonad and brain tissues. Bluehead wrasse *it* and *ef1a* sequences were previously published (Genbank MF279538.1 and MF279537.1, respectively) (*Todd et al., 2017*). Contigs were partially verified using PCR. PCR primers were designed against the contig sequence for each gene in bluehead and spotty wrasse using Primer3 (*Untergasser et al., 2012*), and are shown in Table S3. Reactions (20 µL) contained 10 ng cDNA, 1×MyTaq reaction buffer (Bioline), 1×MyTaq DNA Polymerase (Bioline), and 0.5 µM forward and reverse primers. Reactions were run in a Mastercycler Pro S thermal cycler (Eppendorf) with the following protocol: 95 °C (3 mins) followed by 35 cycles of 95 °C (30 s), annealing at 5 °C below primer melting temperature (see Table S3) (30 s), and 72 °C (45 s), with a final extension at 72 °C (15 mins). PCR products were visualised by electrophoresis through a 1% agarose gel using SYBR Safe DNA Gel Stain (Invitrogen). Amplicons of expected sizes were gel-extracted using a NucleoSpin gel and PCR clean-up kit (Macherey-Nagel). Extracted products were Sanger sequenced (Genetic Analysis Services, Department of Anatomy, University of Otago) in both directions using the respective PCR primers. For kyusen wrasse, bluehead wrasse PCR primers were used to determine partial gene sequences. Forward and reverse amplicons were aligned to create a consensus kyusen sequence for each gene in Geneious R10 (*Kearse et al., 2012*). Primers for *amh* and *it* did not amplify kyusen DNA, however qPCR primers designed for bluehead wrasse were successful in kyusen (see Table S4).

## Quantitative real-time PCR

For each gene, species-specific primers were designed nested within the verified partial gene sequences in Primer3 (see Table S4). Primers were designed to cross exon-exon boundaries to avoid amplifying residual contaminating DNA.

Quantitative real-time PCR (qPCR) was used to measure mRNA levels for each gene in either gonad or brain using the QuantStudio 5 Real-Time PCR system (ThermoFisher). All samples, including standards and negative controls, were run in duplicate (bluehead

wrasse) or triplicate (all other samples) in a 96-well plate. An inter-plate calibrator (cDNA from 6 randomly chosen individuals) was run in triplicate for each spotty qPCR assay. Target gene DNA previously obtained by PCR was used to create standard curves consisting of seven 10-fold dilutions. Reactions (10 µL) contained 20 ng cDNA (except for *18S*, 0.2 ng), 1 µM forward and reverse primers, 1×SYBR® Premix Ex Taq™ II (Tli RNaseH Plus) (Takara), and 1×ROX reference dye (Takara). Bluehead wrasse samples were run without ROX reference dye. Cycling conditions were 95° C (2 mins) followed by 40 cycles of 95° C (5 s), annealing temperature (see Table S4) (10 s), and 72° C (5 s). Melt curve analysis was run to verify the production of a single product which was then confirmation-sequenced (Genetic Analysis Services, Department of Anatomy, University of Otago). Further qPCR details are supplied in a MIQE table (see Table S5).

## Statistical analysis

Due to non-normality of the raw qPCR data, the non-parametric Kruskal-Wallis test was used, followed by *post hoc* comparisons using Dunns tests, with Benjamini Hochberg correction for multiple comparisons, in R (*R Core Team, 2014*) (Data S1 and S2). Expression of *ef1a, 18S* and *g6pd* as well as the geometric mean of all possible combinations, was tested for use as reference genes. Data were normalised using reference gene(s) whose expression was not significantly affected by sex and that showed the flattest expression profile across sexes (see Table S6 for chosen reference genes). Overall, data normalisation had minimal effect on the trend of the results except for spotty wrasse survey 1, in which normalisation masked otherwise clear sex-specific trends for both gonadal genes (Table S6, Figs. S1E and S1F). Therefore, results are presented for un-normalised data. Results for normalised data are available as supplemental materials (see Fig. S1 for gonadal genes and Fig. S2 for brain genes). For each experiment, graphed results are presented as expression relative to control females (i.e., all other sample quantities are expressed as an n-fold difference relative to the control female group).

## Phylogenetic analysis

Robust fine-scale phylogenies support comparative analyses of labrids (*Baliga & Law, 2016*). However, as these do not yet include the spotty wrasse, we undertook phylogenetic analyses to resolve the relationship of spotty wrasse to the bluehead and kyusen wrasses. Sequences of the 12S and 16S mitochondrial ribosomal genes were produced for all three species, using PCR primers from *Westneat & Alfaro (2005)*, and they were combined with sequences from 296 labrid taxa analysed in *Baliga & Law (2016)* (kindly provided by Dr. Vikram Baliga). Genomic DNA was extracted from ovary (kyusen and bluehead wrasse) and liver (spotty) samples using a standard lithium-chloride protocol (*Gemmell & Akiyama, 1996*). Mitochondrial ribosomal genes *12S* and *16S* were PCR-amplified using reactions (20 µL) containing 10 ng DNA, 1×NH$_4$ reaction buffer (Bioline), 1×BIOTAQ DNA Polymerase (Bioline), 2mM MgCl$_2$ solution, 1mM dNTP mix, and 1 µM forward and reverse primers. Reactions were run in a Sure Cycler 8800 (Agilent Technologies) with the following protocol: 94 °C (2 mins) followed by 30 cycles of 94 °C (30 s), 60 °C (12S) or 49 °C (16S) (30 s), and 72 °C (55 s), with a final extension at 72 °C (2 mins). PCR

products were visualised by electrophoresis through a 1% agarose gel using SYBR Safe DNA Gel Stain (Invitrogen), purified using AcroPrep Advance 96-well filter plates (Pall Corporation), and Sanger sequenced in both directions using the respective PCR primers (Genetic Analysis Services, Department of Anatomy, University of Otago).

Phylogenetic relationships within the Labridae were reconstructed using Bayesian inference in MrBayes 3.2.6 (*Ronquist & Huelsenbeck, 2003*), using the CIPRES Science Gateway v3.3. *12S* and *16S* sequences were concatenated following determination of the best-fit model of nucleotide substitution for each gene (GTR + I + G, based on AIC, BIC and DT scores) in jModelTest 2.0 (*Darriba et al., 2012*) (Data S3). A partitioned analysis was carried out with four separate runs, each from a different random starting tree. Default settings were used as priors, and four Markov chains were sampled every 10, 000 generations over 71.2 million Markov chain Monte Carlo generations. Convergence was supported by the average standard deviation of split frequencies of independent runs falling below 0.01. Bayesian posterior probabilities were calculated after discarding the first 25% of sampled trees burn-in. The 50% majority rule consensus tree was prepared in FigTree v1.4.3 (http://tree.bio.ed.ac.uk/).

## RESULTS

### Labridae phylogeny
Spotty wrasse was placed with strong statistical support (>90% bootstrap support) within the Pseudolabrines, together with other *Notolabrus* spp (Fig. 2). This group is resolved as sister to the Labrichthyines and Julidines, which contains the bluehead and kyusen wrasse. Our analysis places the Labrichthyines as sister to the Julidines, as in previous labrid phylogenies (*Westneat & Alfaro, 2005*; *Cowman, Bellwood & Van Herwerden, 2009*), whereas the *Baliga & Law (2016)* topology positions the Labrichthyines within the Julidines.

### Sex change
#### Experiment 1: social induction of sex change in wild bluehead wrasse
Social manipulations successfully induced sex change in wild female bluehead wrasses, and form part of whole-transcriptome analyses described elsewhere (*Todd et al., 2017*; *Liu et al., 2015*). Three samples representative of each sex change stage, plus control females, TP and IP males were analysed herein (Fig. 3).

#### Experiment 2: social induction of sex change in captive spotty wrasse
Removal of TP males readily induced sex change in captive female spotty wrasse (Figs. 4 and 5). Histological analysis revealed that in the manipulated tanks (TP male removed), 15 fish reached ET stage (day 30 $n = 2$, day 50 $n = 3$, day 60 $n = 4$, day 65 $n = 3$, day 66 $n = 3$), one reached MT stage (day 50), one LT stage (day 50), and one was classified as fully TP male (day 60). Only four females within the manipulated tanks showed no histological signs of sex change upon sampling (day 30 $n = 2$, day 66 $n = 2$). There was no conclusive evidence of sex change by females in control tanks (TP male present). However, four control females (day 30 $n = 3$, day 66 $n = 1$) showed evidence of early ovarian atresia indicative of an ET stage, although this may represent normal atresia following the breeding season. Across the

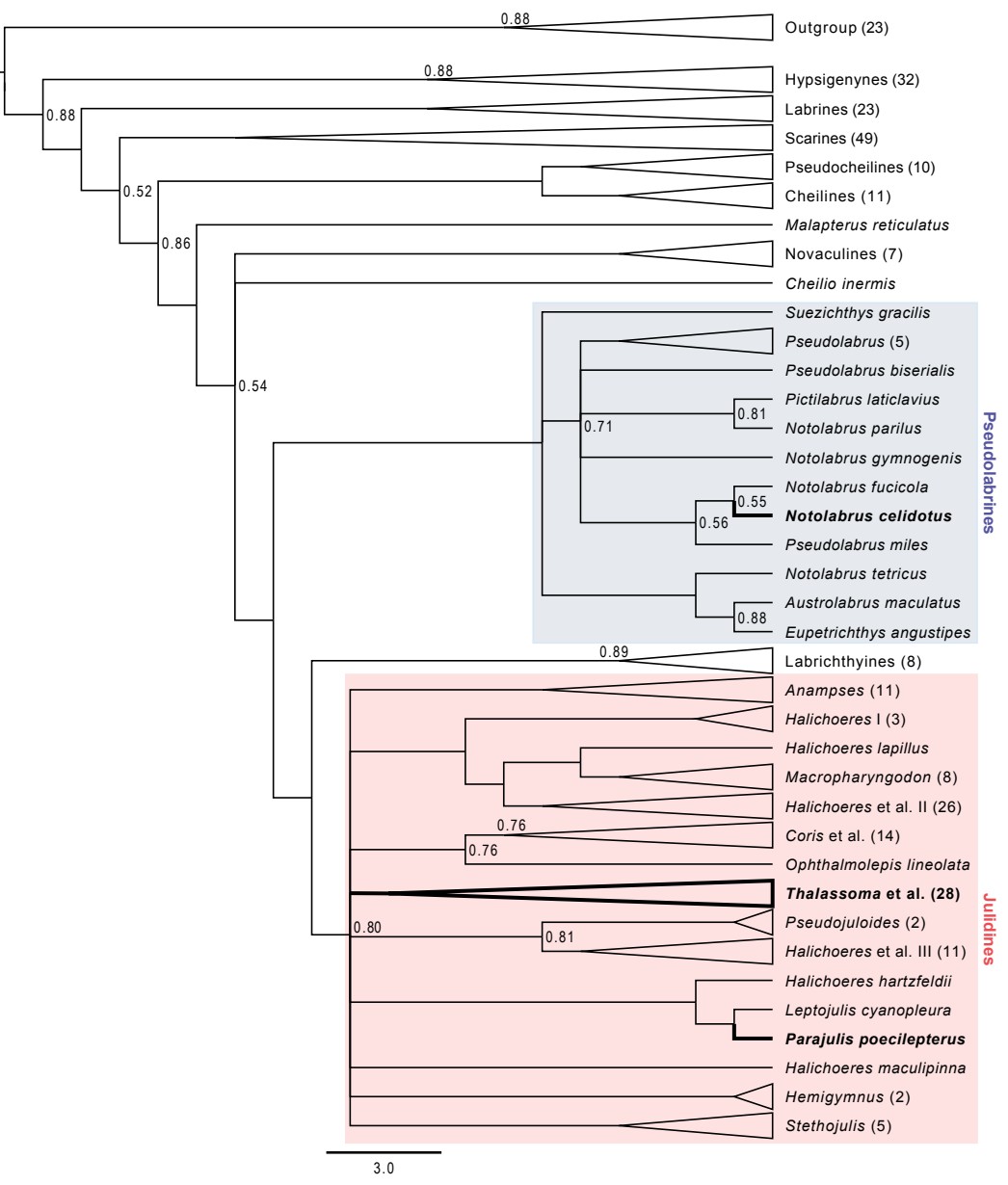

**Figure 2  Majority rule consensus tree from Bayesian MCMC analyses.** The tree is simplified to show relationships between the bluehead, spotty, and kyusen wrasses. Unlabelled nodes have Bayesian posterior probabilities >0.90. Tip labels are the species or genus names, with the number of species sampled in brackets. A triangular tip indicates the clade has been collapsed.

entire experiment, five of the original 53 IP fish were found to be IP males after histological analysis.

### Survey 1: opportunistic sampling of spotty wrasse

Among the opportunistically caught spotty wrasse, fish were found at a range of stages (NBF $n = 6$, ET $n = 3$, MT $n = 2$, LT $n = 2$, TP male $n = 1$) (Fig. 5).

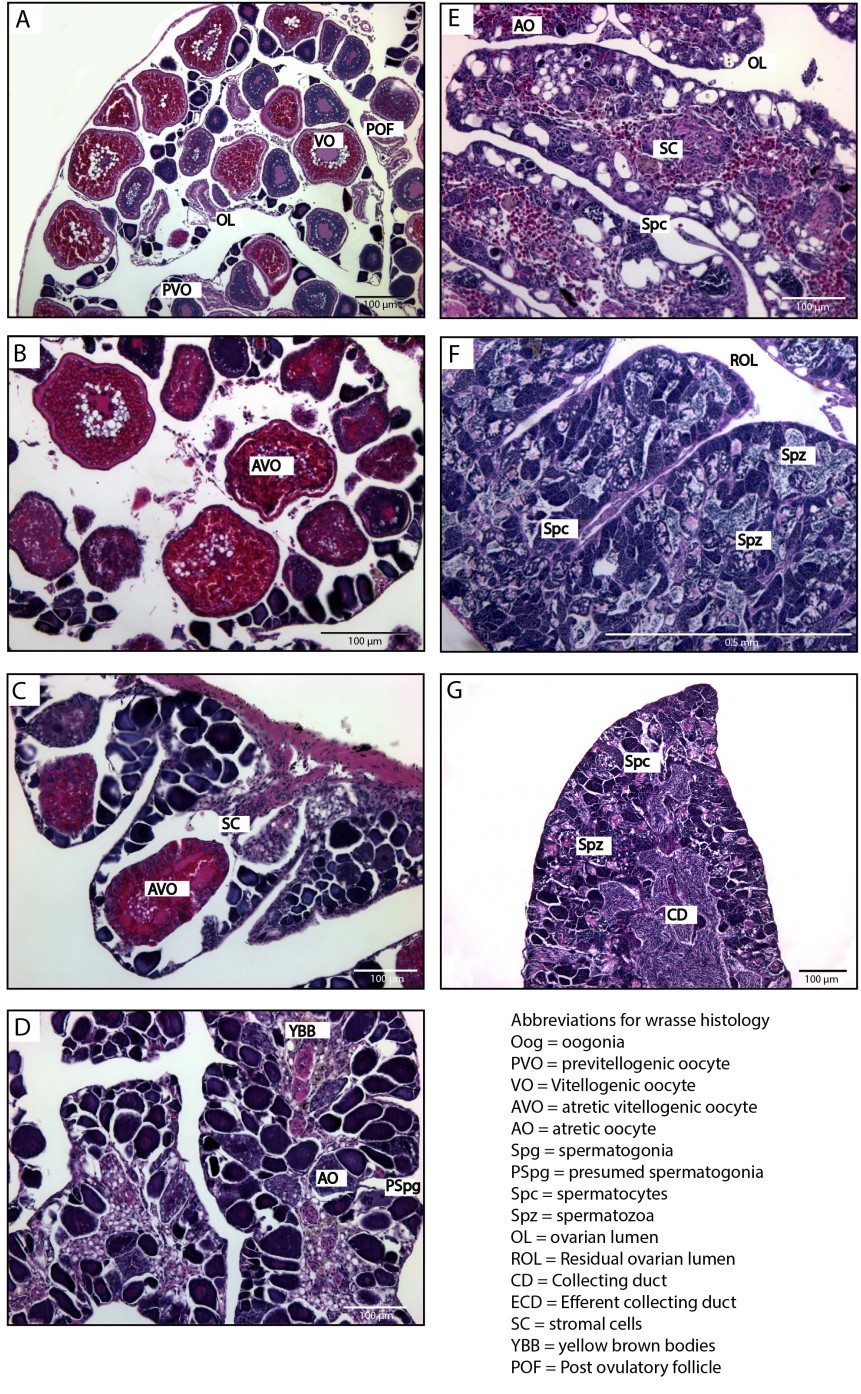

Abbreviations for wrasse histology
Oog = oogonia
PVO = previtellogenic oocyte
VO = Vitellogenic oocyte
AVO = atretic vitellogenic oocyte
AO = atretic oocyte
Spg = spermatogonia
PSpg = presumed spermatogonia
Spc = spermatocytes
Spz = spermatozoa
OL = ovarian lumen
ROL = Residual ovarian lumen
CD = Collecting duct
ECD = Efferent collecting duct
SC = stromal cells
YBB = yellow brown bodies
POF = Post ovulatory follicle

**Figure 3** **Histological stages of gonadal sex change in the bluehead wrasse.** (A) Stage 1, breeding female with mature ovary containing pre-vitellogenic and vitellogenic oocytes. (B) Stage 2, atresia of vitellogenic oocytes. (C) Stage 3, atresia of pre-vitellogenic and vitellogenic oocytes and clustering of stromal cells. (D) Stage 4, proliferation of presumed spermatogonia. (E) Stage 5, spermatogenesis begins. (F) Stage 6, mature testis with spermatozoa and a residual ovarian lumen. (G) Initial phase male containing spermatozoa, where absence of a residual ovarian lumen suggests this fish has not sex changed and is an initial phase male. Scale bar, 100 μm (A, B, C, D, E, G), 0.5 mm (F). Stages follow the classification of *Nakamura et al. (1989)*.

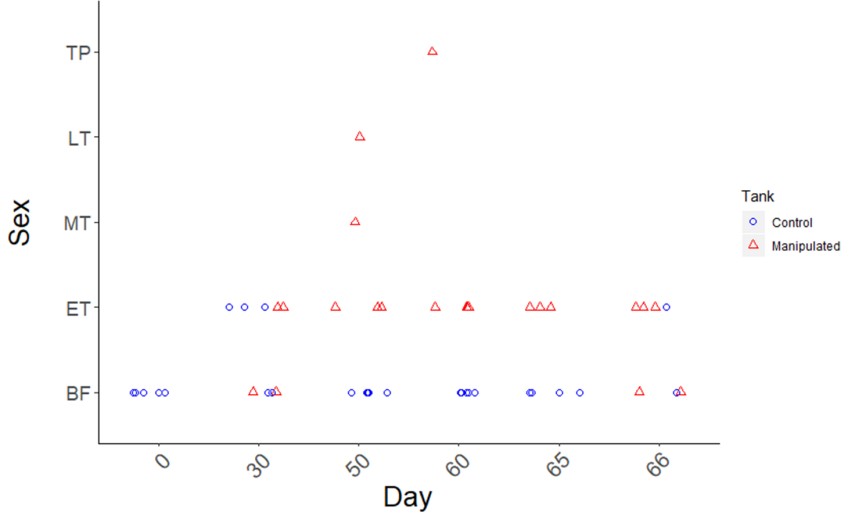

**Figure 4** **Time course of sex change in the spotty wrasse following social manipulation in captivity (Experiment 2).** Points represent the sex change stage of each sampled female on each sampling day. Blue circles are samples from control tanks (TP male present - non-permissive environment), and red triangles from manipulated tanks (TP male removed - permissive environment).

### Survey 2: wild-caught kyusen wrasse

Wild-caught kyusen wrasse were mostly females (NBF $n = 7$) and TP males (TP $n = 11$), plus a few ET females ($n = 3$) and IP males ($n = 3$) (Fig. 6). The ET fish were sampled in late September.

## Quantitative real-time PCR
### Gonad: cyp19a1a

In all three species, *cyp19a1a* expression was highest in ovaries and near-zero in TP and IP male testes (Fig. 7). In bluehead and spotty wrasses, *cyp19a1a* expression decreased across progressive sex change stages.

### Experiment 1: social induction of sex change in wild bluehead wrasse

Sex had a significant effect on *cyp19a1a* expression ($X^2(8) = 25.08$, $p < 0.01$). At the onset of behavioural sex change (stage 1), a spike in *cyp19a1a* expression occurred (median 1.8 fold higher than control females (CF)), followed by near zero expression from stage 2 onwards (onset of ovarian atresia). There was a significant difference in the distribution of *cyp19a1a* expression between control females and TP (median 0.0001-fold that of CF, $p < 0.05$) and IP males (median 0.00007-fold that of CF, $p < 0.05$), but not between control females and stages 1–6. However, there was a clear trend of decreasing expression (Fig. 7).

### Experiment 2: social induction of sex change in captive spotty wrasse

Sex significantly affected *cyp19a1a* expression ($X^2(7) = 38.35$, $p < 0.0001$). Decreased *cyp19a1a* expression was first observed among females at the ET stage (median 0.3-fold lower than C BF D0, non-significant; median 0.28-fold lower than C BF, $p < 0.01$). The

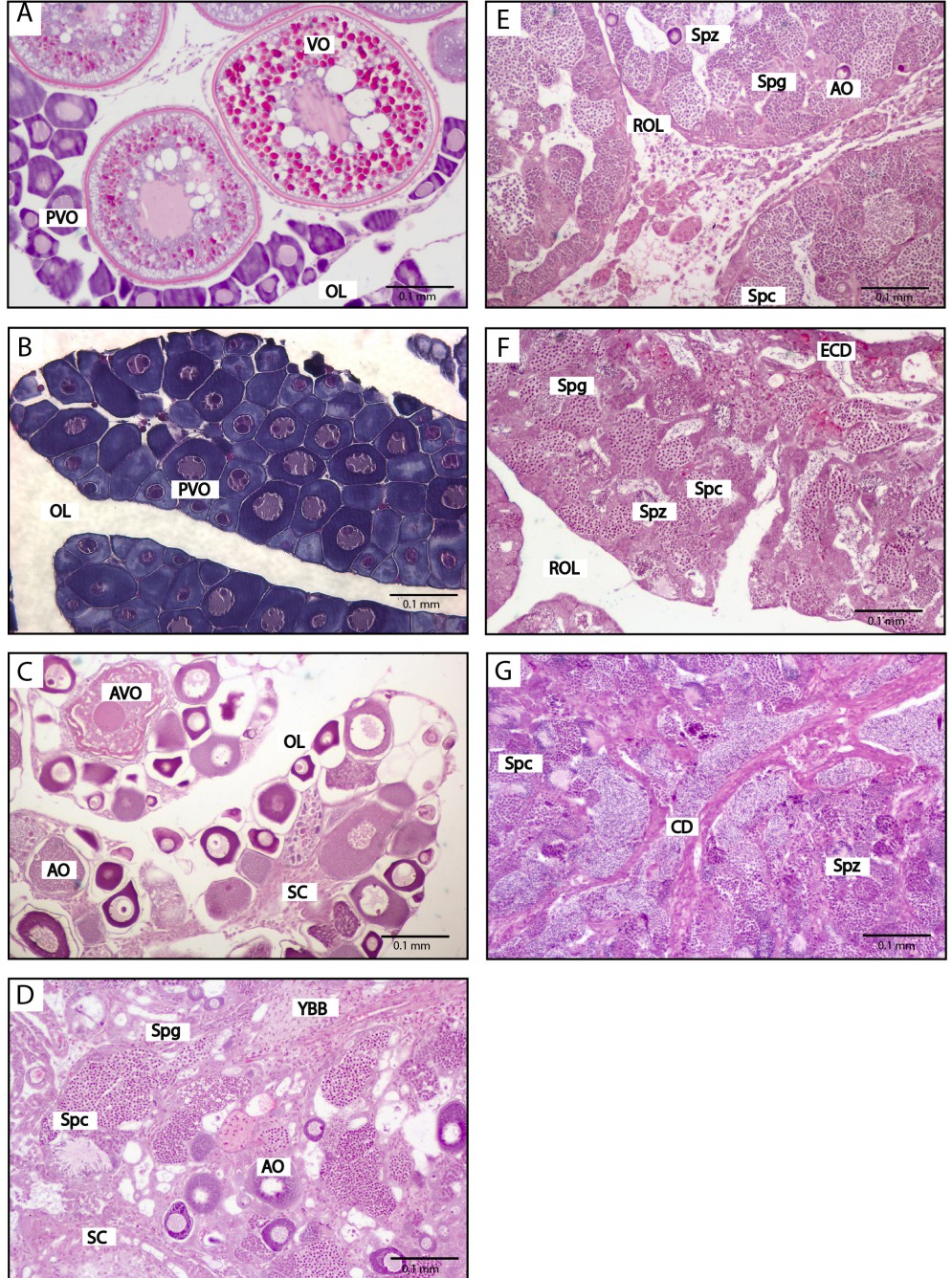

**Figure 5  Histological stages of gonadal sex change in the spotty wrasse.** (A) Breeding female with pre-vitellogenic and vitellogenic oocytes. (B) Non-breeding female predominated by pre-vitellogenic oocytes. (C) Early transitional; atresia of oocytes and presence of stromal cells. (D) Mid transitional; oocyte numbers diminished and ovarian follicles were largely atretic, with proliferation of spermatogonia. (E) Late transitional; spermatogenic cysts predominate over atretic oocytes. (F) Terminal phase male; mature testis with spermatozoa in cysts arranged into seminiferous tubules with presence of a residual ovarian lumen. (G) Initial phase male containing spermatozoa, where absence of a residual ovarian lumen suggests this fish has not sex changed and is an initial phase male. Scale bar, 0.1 mm. See Fig. 3 for abbreviations.

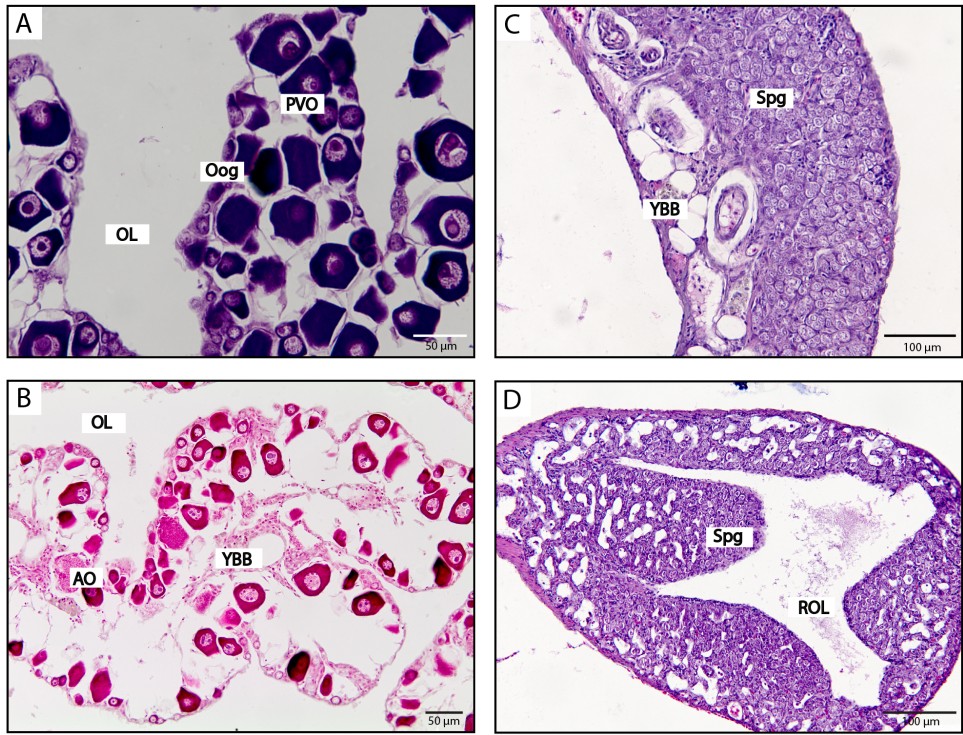

**Figure 6** **Histological stages of gonadal sex change in the kyusen wrasse.** (A) Non-breeding female with pre-vitellogenic oocytes. (B) Early transitional; atretic oocytes and yellow brown bodies. (C) Late transitional; proliferation of spermatogonia. (D) Terminal phase male; mature testis with lobular structure and a residual ovarian lumen. Scale bar, 100 μm (A, B), 50 μm (C, D). See Fig. 3 for abbreviations.

single MT fish had a median *cyp19a1a* expression 0.25-fold that of C BF D0, while the single LT individual had near-zero *cyp19a1a* expression (median 0.005-fold that of C BF D0). Distribution of gonadal *cyp19a1a* expression was significantly reduced in TP and IP male testes compared with ovaries of all control females (median in both males 0.02-fold that of C BF D0, $p < 0.01$).

### Survey 1: opportunistic sampling of spotty wrasse
Sex did not have a significant effect on *cyp19a1a* expression ($X^2(4) = 9.23$, $p = 0.06$). However, a clear trend was observed with *cyp19a1a* expression at near-zero levels in MT (median 0.004-fold that of NBFs) and LT stage fish (median 0.03-fold that of NBFs), and in the single TP male (median 0.003-fold that of NBFs). Gonadal *cyp19a1a* expression in three ET samples ranged from 0.3 to 4.3-fold higher than that seen among the NBFs.

### Survey 2: wild-caught kyusen wrasse
Sex significantly affected *cyp19a1a* expression ($X^2(3) = 17.02$, $p < 0.001$). There was no difference in *cyp19a1a* expression between NBFs and the three samples staged as ET. Although *cyp19a1a* expression was near-zero in TP and IP males (both median 0.1-fold that of NBF), only TP males showed a significant difference in distribution compared with females ($p < 0.01$).

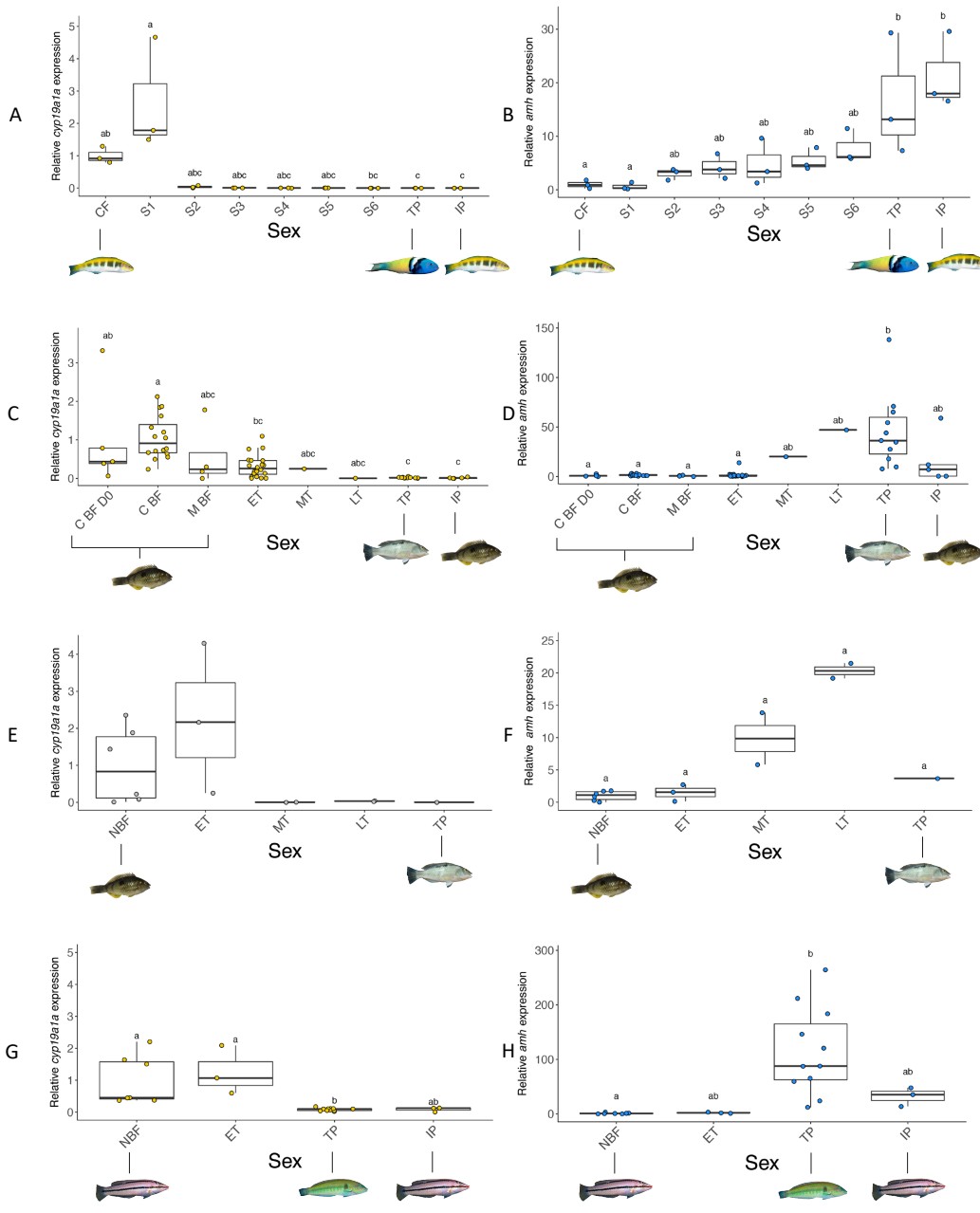

**Figure 7  Relative gonadal expression of *cyp19a1a* (A, C, E, G) and *amh* (B, D, F, H) mRNA.** Expression levels are compared among females, transitioning fish, TP males and IP males. (A, B) Bluehead wrasse induced to change sex in the wild (Experiment 1). (C, D) Spotty wrasse induced to change sex in captivity (Experiment 2). (E, F) Wild-caught spotty wrasse (Survey 1). (G, H) Wild-caught kyusen wrasse (Survey 2). Points represent individual fish. Boxplots represent the median, lower and upper quartile values, and 1.5-fold the interquartile range. Yellow, blue and grey points indicate expression is significantly female-biased, male-biased, and non-significantly different, respectively. Letters denote 

**Figure 7 (…continued)**
a significant difference in distribution between groups and 'a' indicates overall significance without significant pairwise differences. Sample sizes: bluehead wrasse $n = 3$, all groups; spotty wrasse socially induced to change sex in captivity C BF D0 $n = 5$, C BF $n = 16$, M BF $n = 4$, ET $n = 20$, MT $n = 1$, LT $n = 1$, TP $n = 11$, IP $n = 5$; spotty wrasse opportunistically caught NBF $n = 6$, ET $n = 3$, MT $n = 2$, LT n $= 2$, TP $n = 1$; kyusen wrasse NBF $n = 7$, ET $n = 3$, TP $n = 11$, IP $n = 3$. Abbreviations: C BF D0, breeding female from control tank (TP male present) at experiment day 0; C BF, breeding female from control tank (TP male present) removed at progressive time points throughout the experiment; CF, control female; ET, early transitional; IP, initial phase male; LT, late transitional; M BF, breeding female from manipulated tanks (TP male removed) removed at progressive, time points throughout experiment; MT, mid transitional; NBF non-breeding female; S1-6, stages 1-6; TP, terminal phase male. See Fig. 1 legend for photo credits of female and male bluehead, spotty and kyusen wrasses.

### Gonad: amh

Gonadal *amh* expression showed a pattern opposite to that of *cyp19a1a*. In all three species, *amh* expression was near-zero in females and highest in TP males, with a clear trend of increasing expression across sex change (Fig. 7). In all experiments, sex had a significant effect on *amh* expression (bluehead wrasse: $X^2(8) = 21.46$, $p < 0.01$, spotty wrasse experiment: $X^2(7) = 33.18$, $p < 0.0001$, spotty wrasse survey: $X^2(4) = 9.63$, $p < 0.05$, kyusen wrasse: $X^2(3) = 18.46$, $p < 0.001$).

### Experiment 1: social induction of sex change in wild bluehead wrasse

Increased *amh* expression was obvious from stage 2 (median 3-fold higher than CF), and steadily increased to a significantly higher distribution in TP (median 13-fold higher than CF, $p < 0.05$) and IP males (median 18-fold higher than CF, $p < 0.05$).

### Experiment 2: social induction of sex change in captive spotty wrasse

There was a trend of progressive *amh* upregulation beginning in the single fish staged as MT (20-fold higher than C BF D0), continuing in the LT individual (47-fold higher than C BF D0) and reaching a significantly higher distribution in TP males (median 36-fold higher than C BF D0, $p < 0.01$). IP males showed a distribution of *amh* expression intermediate to that of all fish with an intact ovary, and TP males (median 7.23-fold higher than C BF D0).

### Survey 1: opportunistic sampling of spotty wrasse

Despite a significant effect of sex on *amh* expression, *post hoc* analysis showed no significant differences between individual sex stages. ET fishes showed similar expression levels to NBFs, while MT and LT stage fish showed a trend of increased *amh* expression (median 10-fold and 20-fold higher than NBFs, respectively). Expression of *amh* in the single TP male was 3.7-fold higher than in NBFs.

### Survey 2: wild-caught kyusen wrasse

ET fish showed similar *amh* expression to NBFs (median 2-fold higher). TP males had a significantly higher distribution of *amh* expression (88-fold higher than NBF, $p < 0.001$), while *amh* mRNA levels in IP male were intermediate to those of NBFs and TP males (35-fold higher than NBFs).

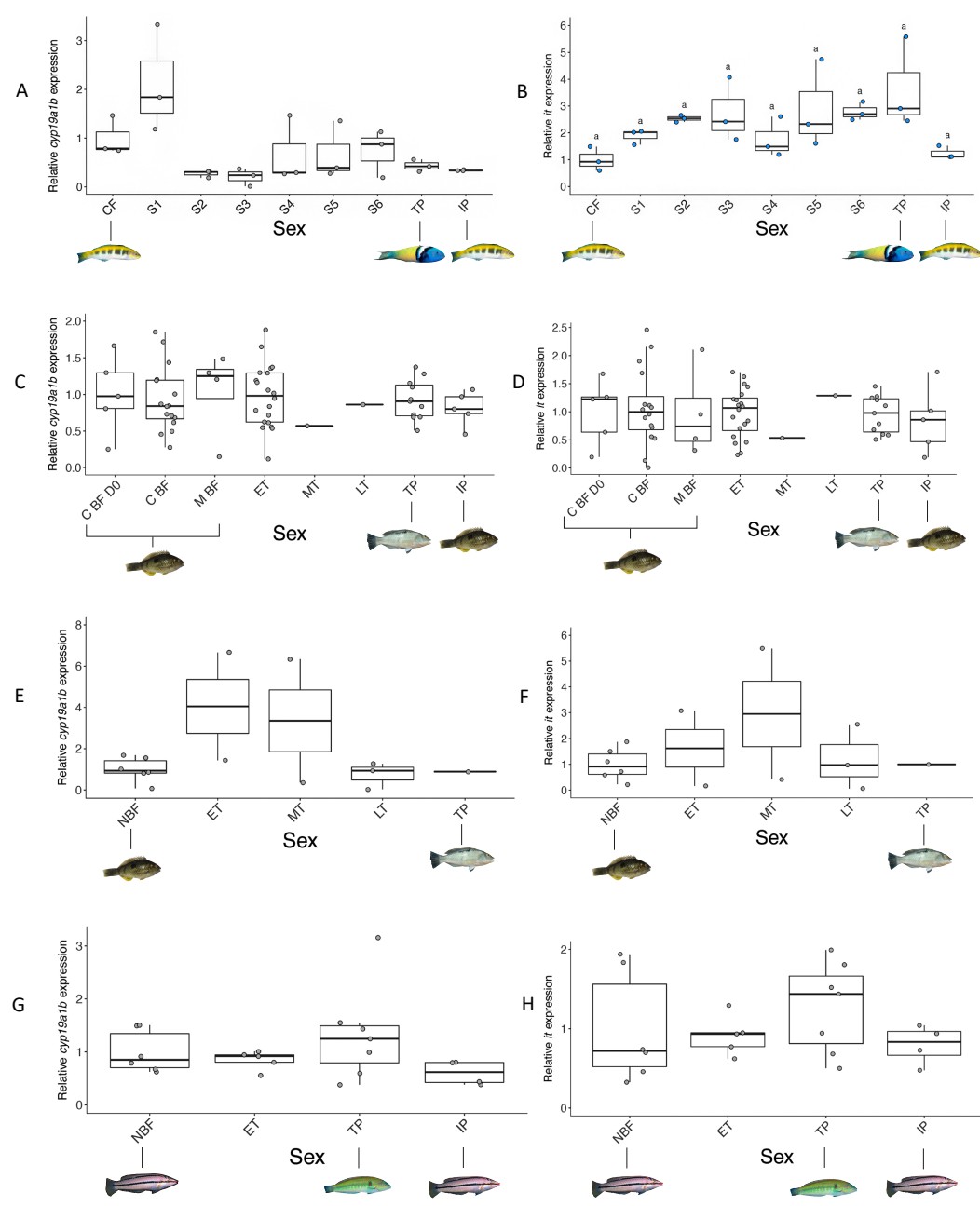

**Figure 8** **Relative brain expression of *cyp19a1b* (A, C, E, G) and *it* (B, D, F, H) mRNA.** Expression levels are compared among females, transitioning fish, TP males and IP males. (A, B) Bluehead wrasse induced to change sex in the wild (Experiment 1). (C, D) Spotty wrasse induced to change sex in captivity (Experiment 2). (E, F) Wild-caught spotty wrasse (Survey 1). (G, H) Wild-caught kyusen wrasse (Survey 2). Points represent individual fish. Boxplots represent the median, lower and upper quartile values, and 1.5-fold the interquartile range. Yellow, blue and grey points indicate expression is significantly female-biased, male-biased, and non-significantly different, respectively. Letters denote a significant difference in distribution between groups and 'a' indicates overall significance without significant pairwise differences. Sample sizes: bluehead wrasse *n* = 3 all groups; spotty wrasse socially induced (continued on next page...)

### Brain: cyp19a1b

For all three species, sex did not have a significant effect on *cyp19a1b* expression (bluehead wrasse: $X^2(8) = 13.64$, $p = 0.09$, spotty experiment: $X^2(7) = 2.88$, $p = 0.90$, spotty survey: $X^2(4) = 3.13$, $p = 0.54$, kyusen wrasse: $X^2(3) = 3.00$, $p = 0.28$) (Fig. 8). However, in the bluehead wrasse a subtle trend is evident similar to that of *cyp19a1a* expression in the gonad; expression peaks transiently at stage 1 (median 1.8 fold higher than CF), then decreases at stage 2 (median 0.3-fold that of CF), remaining at low levels at subsequent sex change stages and in TP (median 0.4-fold that of CF) and IP males (median 0.3-fold that of CF). No clear trends in *cyp19a1b* expression were evident in brain samples from spotty or kyusen wrasses.

### Brain: it

In bluehead wrasse, sex did have a significant effect on *it* expression ($X^2(8) = 18.12$, $p < 0.05$). However, *post hoc* analysis showed no significant differences between individual sex stages. There was a trend of increasing *it* expression in fore/midbrain of bluehead wrasse, beginning at stage 1 (median 2-fold higher than CF) and progressively increasing to highest levels in TP males (median 3-fold higher than CF). IP male *it* expression was similar to that of control females (median 1.1-fold higher than CF). In spotty and kyusen wrasses, sex did not have a significant effect on *it* expression (spotty experiment: $X^2(7) = 2.82$, $p = 0.90$, spotty survey: $X^2(4) = 4.00$, $p = 0.98$, kyusen wrasse: $X^2(3) = 1.87$, $p = 0.60$), nor were there any clear trends.

## DISCUSSION

In order to understand sex change from a functional and evolutionary standpoint, an important question is to what degree genetic systems regulating sex change are conserved or different among species. Using a comparative qPCR approach across three wrasse species which share protogyny as an ancestral state, we investigated the roles of *cyp19a1a* and *amh* as proximal regulators of gonadal sex change, and *cyp19a1b* and *it* as regulators of behavioural sex change in the brain. We evaluate whether these genes may form part of a conserved molecular machinery underlying protogynous sex change in wrasses, and whether any differences exist among tropical and temperate species differing in the seasonality of sex change and the rigidity of social hierarchies.

## Gonadal sex change—*cyp19a1a* and *amh* as proximal regulators

In protogynous hermaphrodites, interrupted *cyp19a1a* expression has been suggested as the molecular switch that initiates ovarian atresia and gonadal sex change (*Todd et al., 2016*; *Liu et al., 2016*). Experimental studies have shown that treatment of adult females with aromatase inhibitors can induce complete sex change, both in year-round (*Higa et al., 2003*; *Nozu, Kojima & Nakamura, 2009*) and seasonal (*Bhandari et al., 2005*; *Kroon & Liley, 2000*; *Li et al., 2006*) breeders. However, whether *cyp19a1a* downregulation acts as a proximal switch initiating natural gonadal sex change broadly in protogynous species has been unclear. Firstly, because prior studies in other species have not examined atretic ovaries from females during earliest sex change, i.e., before proliferation of male tissues, they could not confirm whether *cyp19a1a* downregulation occurs coincidentally with the initiation of gonadal sex change in year-round (*Zhang et al., 2008*; *Liu, Guiguen & Liu, 2009*) and seasonal (*Li et al., 2006*; *Huang et al., 2009*) breeders. Secondly, in seasonally-breeding species where sex change proceeds from an already regressed ovary with lower oestrogen production and aromatase activity (*Li, Liu & Lin, 2007*), *cyp19a1a* downregulation may be less important.

In the tropical bluehead wrasse, *cyp19a1a* expression dropped to near zero at the first signs of ovarian atresia before the presence of male tissues (Fig. 7A). Our spotty and kyusen wrasse samples include early transitioning females with advanced ovarian atresia prior to the appearance of male tissues (ET stage). In spotty wrasse socially manipulated to change sex in captivity, *cyp19a1a* mRNA levels in ET ovaries are intermediate to those of control breeding females and males (Fig. 7C). In both species caught from the wild, *cyp19a1a* mRNA levels in ET ovaries are within the range recorded for non-breeding females (Figs. 7E and 7G). Therefore, in these temperate species *cyp19a1a* expression is already low prior to sex change. This is expected, given the seasonal ovarian atresia that occurs in temperate protogynous hermaphrodites following the breeding season, and that precedes sex change (*Li, Liu & Lin, 2007*). Overall, our results confirm that suppression of *cyp19a1a* expression is a prerequisite feature of ovaries prior to sex change in protogynous wrasses. Although *cyp19a1a* downregulation may act as a proximal trigger for sex change in tropical year-round breeders like the bluehead wrasse, already low *cyp19a1a* expression in non-breeding ovaries of temperate species may prime them for sex change. As a result, for temperate wrasses that breed seasonally, reduced aromatase expression is not a conclusive marker to distinguish early stage sex changers from non-breeding females with atretic ovaries. Due to this seasonal atresia, care should also be taken when delineating early sex changers and non-breeding individuals with atretic ovaries.

Amh is an important regulator of male phenotype in fishes, regulating germ cell proliferation and differentiation in the testis of numerous species (*Pfennig, Standke & Gutzeit, 2015*; *Schulz et al., 2010*; *Pala et al., 2008*; *Miura et al., 2002*). Our data indicate that *amh* is upregulated during sex change with significantly greater *amh* expression evident in TP males compared to that of females in all three wrasses (Figs. 7B, 7D, 7F and 7H). Testicular production of Amh occurs primarily in Sertoli cells surrounding type A undifferentiated spermatogonia, where it suppresses germ cell proliferation and differentiation as well as steroidogenesis in the interstitial Leydig cells (*Pfennig, Standke*

& Gutzeit, 2015; Skaar et al., 2011). As such, *amh* expression may be expected to increase with spermatogonial recruitment during sex change. In tropical wrasses, spermatogonia proliferate from stage 4 (*Nakamura et al., 1989*) and in a recent study of three-spot wrasse (*Halichoeres trimaculatus*) induced to change sex via aromatase inhibition, *amh* was first upregulated in stage 4 gonads (*Horiguchi et al., 2018*). In the current study, *amh* was also upregulated concurrently with the appearance of male tissues in temperate spotty wrasse (MT stage, equivalent to stage 4) (Figs. 7D and 7F), although timing was inconclusive in kyusen wrasse due to the lack of MT samples (Fig. 7H). By contrast, in bluehead wrasse *amh* was clearly upregulated prior to the appearance of male cells, at the first sign of ovarian atresia (stage 2, Fig. 7B). Together, these data strongly suggest *amh* promotes testicular formation and spermatogenesis during protogynous sex change in wrasses, but also implicate *amh* in initiating maleness in some wrasse species.

In all three wrasses, *amh* was upregulated coincidentally with the downregulation of *cyp19a1a* at early sex change from IP female to TP male (Fig. 7). An inverse relationship between *amh* and *cyp19a1a* expression is widely reported in fishes (*Pfennig, Standke & Gutzeit, 2015*), including zebrafish (*Danio rerio*) (*Rodríguez-Marí et al., 2005*; *Wang & Orban, 2007*), Japanese flounder (*Paralichthys olivaceus*) (*Kitano et al., 2007*), pejerrey (*Odontesthes bonariensis*) (*Fernandino et al., 2008*), Southern flounder (*Paralichthys lethostigma*) (*Mankiewicz et al., 2013*) and rainbow trout (*Oncorhynchus mykiss*) (*Vizziano et al., 2008*). Together, these data suggest a bidirectional antagonism between *amh* and *cyp19a1a* may operate to control sexual fate in fishes (*Todd et al., 2016*), presumably acting within a broader antagonistic framework between core feminising (e.g., *cyp19a1a*, *foxl2*, *wnt4*) and masculinising (e.g., *dmrt1*, *sox9*, *amh*) gene networks known to be responsible for directing and maintaining sexual fate in vertebrates (*Herpin & Schartl, 2011b*).

## Aromatase and isotocin in the brain

In species where sex change is socially cued, complex neurochemical changes in the brain presumably translate visual social information into behavioural and reproductive responses necessary for sex change. Prior work has identified several neuropeptides as likely regulators of behavioural sex change in social tropical wrasses, including arginine vasotocin, Cyp19a1b, It, and gonadotropin-releasing hormone (reviewed in *Lamm et al., 2015*; *Larson, 2010*; *Godwin, 2010*).

Our data do not strongly support a prominent role for *cyp19a1b* in initiating behavioural sex change in protogynous wrasses. In spotty and kyusen wrasses, *cyp19a1b* expression was neither sex-specific nor showed any clear trend across sex change. In bluehead wrasse, although *cyp19a1b* expression clearly decreased with sex change, the trend was non-significant and only noticeable from stage 2, after behavioural changes first occur at stage 1. Likewise, *Black et al. (2011)* found whole brain aromatase activity declined only after behavioural changes in female-to-male sex change of the bluebanded goby (*Lythrypnus dalli*). Expression of *cyp19a1b* in bluehead wrasse fore/midbrain closely parallels gonadal *cyp19a1a* mRNA levels, and may reflect peripheral changes in E2 via putative oestrogen response elements in the *cyp19a1b* promotor (*Diotel et al., 2010*). However, exogenous E2 treatment of bluehead wrasse stimulated brain *cyp19a1b* expression and prevented

behavioural sex change under socially permissive conditions (*Marsh-Hunkin et al., 2013*). Brain gene expression patterns are highly heterogeneous and it remains possible that localised *cyp19a1b* expression changes are important, but would not be detected in studies at a whole brain or fore/midbrain level.

Our data supports the role isotocin plays in modulating teleost socio-sexual behaviours, and social dominance hierarchies in particular (*Godwin & Thompson, 2012*; *Lema, Sanders & Walti, 2015*; *Almeida et al., 2012*). In bluehead wrasse, *it* expression is TP male biased and upregulated across sex change, beginning with a median 2-fold increase at stage 1 when behavioural changes first occur. In tropical wrasses, stage 1 is characterised by rapid (minutes to hours) increases in aggression and male-typical courtship behaviours in transitioning females (*Warner & Swearer, 1991*) that are presumably critical for establishing dominance as the new TP male before gonadal sex change ensues. In wrasses generally, IP males mimic female behaviour and colouration to avoid TP male aggression. We found that *it* expression was as low in bluehead IP males as in females. This further indicates a role for *it* in establishing the dominant TP male phenotype. Our qPCR data validate the same patterns reported in recent whole-transcriptome analyses in this species (*Todd et al., 2017*). An opposite pattern for *it* was observed in the bluebanded goby, a bidirectional hermaphrodite in which high social status is also a critical cue for female-male sex change, yet the number of isotocin-immunoreactive cells in the pre-optic area decreased across female-male sex change (*Black, Reavis & Grober, 2004*). These data and studies in social cichlids indicate isotocin can have species-specific and context-dependent roles in social behaviour (*Reddon et al., 2017*; *O'connor et al., 2015*; *O'connor et al., 2016*).

Expression of *it* was not associated with sexual phenotype or sex change in spotty or kyusen wrasses. Although our social manipulation experiment provides the first evidence confirming sex change is socially cued in spotty wrasse, behavioural markers of sex change have not been characterised in either species. Overall, our data and those of *Black, Reavis & Grober (2004)* support isotocin as an important proximal mediator of behavioural transitions in sequential hermaphrodites with strict social hierarchies. Further work is necessary to clarify whether *it* also regulates socially-cued sex change in temperate wrasses, and may show seasonal fluctuations.

## CONCLUSIONS

This research investigated whether evolutionarily conserved molecular mechanisms underlie protogynous sex change in wrasses. In this first comparison of candidate gene expression in tropical versus temperate protogynous species, we find both conservation and diversity in the regulatory machinery underlying sex change. Our data support conserved roles for *cyp19a1a* and *amh* as important proximal regulators of gonadal sex change in protogynous wrasses - these genes may act concurrently to orchestrate the ovary-testis transition by controlling ovarian atresia and testicular development, respectively. However, differences in timing of expression changes relative to key histological events of sex change may be species-specific or reflect differences between tropical and temperate species in the seasonality or duration of sex change. In the brain, our data do not support a role for brain

aromatase, *cyp19a1b*, in initiating behavioural sex change, as expression changes for this gene trailed rapid behavioural changes. Brain isotocin expression strongly correlated with TP male-specific behaviours and the rapid behavioural changes characterising the onset of sex change in the bluehead wrasse, but not spotty or kyusen wrasses. Characterising behavioural and molecular markers of sex change in temperate wrasses will be important for understanding how visual social cues are transduced to initiate the sex change cascade. Future work employing macro-dissection of the brain will be important, as our sampling of the whole brain or fore/midbrain may have obscured important region-specific signals. Future manipulative experiments will also be important in determining specific functions of these genes in regulating sex change.

## ACKNOWLEDGEMENTS

We are grateful to Carlos Farias Moraes and Holly Robertson for their invaluable assistance in conducting the captive experiments in spotty wrasse. We thank Bill Tylor, Sidney Gaston Sanchez, Brandon Klapheke, Jeannie Brady, and Alison Lukowsky for support collecting bluehead samples. Robbie McPhee assisted with the preparation of figures. Alexander Goikoetxea provided valuable feedback on an earlier draft. Vikram Baliga generously supplied data for the phylogenetic analyses.

### Funding

This work was supported by the Royal Society of New Zealand Marsden Fund (UOO1308 to Neil J. Gemmell), the Japan Society for the Promotion of Science (L10703 to P Mark Lokman) and the National Science Foundation (1257791 to John R. Godwin and 1257761 to Bill Tyler at Indian River State College, Florida). Jodi Thomas was supported by an Otago School of Medical Science Summer Scholarship. The funders had no role in study design, data collection and analysis, decision to publish, or preparation of the manuscript.

### Grant Disclosures

The following grant information was disclosed by the authors:
Royal Society of New Zealand Marsden Fund: UOO1308.
Japan Society for the Promotion of Science: L10703.
National Science Foundation: 1257791.
Indian River State College: 1257761.
Otago School of Medical Science Summer Scholarship.

### Competing Interests

The authors declare there are no competing interests.

### Author Contributions

- Jodi T. Thomas conceived and designed the experiments, performed the experiments, analyzed the data, prepared figures and/or tables, authored or reviewed drafts of the paper, approved the final draft, wrote the manuscript.

- Erica V. Todd conceived and designed the experiments, analyzed the data, prepared figures and/or tables, authored or reviewed drafts of the paper, approved the final draft, wrote the manuscript.
- Simon Muncaster conceived and designed the experiments, performed the experiments, authored or reviewed drafts of the paper, approved the final draft.
- P Mark Lokman conceived and designed the experiments, performed the experiments, authored or reviewed drafts of the paper, approved the final draft.
- Erin L. Damsteegt performed the experiments, authored or reviewed drafts of the paper, approved the final draft.
- Hui Liu performed the experiments, approved the final draft.
- Kiyoshi Soyano conceived and designed the experiments, performed the experiments, approved the final draft.
- Florence Gléonnec and Melissa S. Lamm performed the experiments, approved the final draft.
- John R. Godwin conceived and designed the experiments, performed the experiments, approved the final draft.
- Neil J. Gemmell conceived and designed the experiments, authored or reviewed drafts of the paper, approved the final draft.

## Animal Ethics

The following information was supplied relating to ethical approvals (i.e., approving body and any reference numbers):

The Institutional Animal Care and Use Committee at North Carolina State University provided approval for Experiment 1 (12-069-0). The New Zealand National Animal Ethics Advisory Committee provided approval for Experiment 2 (2015_02) and Survey 1 (92-10). The Animal Care and Use Committee of the Institute for East China Sea Research, Nagasaki University, Japan, provided approval for Survey 2 (#15-06).

## Field Study Permissions

The following information was supplied relating to field study approvals (i.e., approving body and any reference numbers):

Fish were collected with approval from the New Zealand Ministry of Primary Industries (593-3).

## DNA Deposition

The following information was supplied regarding the deposition of DNA sequences:

Candidate gene and reference gene sequences described here are accessible via GenBank accession numbers: bluehead wrasse cyp19a1a MK252274, amh MK252275, cyp19a1b MK252276, it MF279538.1, g6pd MK252277, ef1a MF279537.1, 18S MK246126, spotty wrasse cyp19a1a MK252278, amh MK252279, cyp19a1b MK252280, it MK252281, g6pd MK252282, ef1a MK252283, 18S MK246127, kyusen wrasse cyp19a1a MK252284, amh MK252285, cyp19a1b MK252286, it MK252287, g6pd MK252288, ef1a MK252289, 18S MK246128.

## Data Availability

R code for statistical analysis of qPCR data is available in Data S1. Raw qPCR measurements are available in Data S2. Raw (.raw) and relative (.rel) expression values are provided in separate sheets for each species, and for brain and gonad analyses separately. The sequence alignment used for phylogenetic analyses is available in Data S3. The file contains a Nexus format alignment of concatenated 12S and 16S ribosomal RNA sequence data.

## Supplemental Information

Supplemental information for this article can be found online at http://dx.doi.org/10.7717/peerj.7032#supplemental-information.

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
