# Peer review of "Conservation and diversity in expression of candidate genes regulating socially-induced female-male sex change in wrasses"

_PeerJ, doi:10.7717/peerj.7032_

## Round 0.1 · original submission · Major Revisions

Dear Dr. Thomas and colleagues:

Thanks for submitting your manuscript to PeerJ. I have now received two independent reviews of your work, and as you will see, the reviewers raised some concerns about the research. Despite this, these reviewers are optimistic about your work and the potential impact it will have on research communities studying factors governing female-male sex change in wrasses, as well as reproductive biology in general. Thus, I encourage you to revise your manuscript accordingly, taking into account all of the concerns raised by both reviewers.

Please note: while most of the concerns of the reviewers are relatively minor, this is a major revision to ensure that the original reviewers have a chance to evaluate your responses to their concerns.

I look forward to seeing your revision, and thanks again for submitting your work to PeerJ.

Good luck with your revision,

-joe

Reviewer 1 ·

Basic reporting

no comment

Experimental design

no comment

Validity of the findings

no comment

Additional comments

General comments:
In this manuscript, authors addressed whether degree to which a conserved genetic machinery orchestrates sex change by comparing expression patterns of four candidate regulatory genes among protogynous three species of wrasses. Based on the qOCR results, authors suggested that conserved roles for cyp19a1a and amh as important proximal regulators of gonadal sex change in protogynous wrasses. On the other hand, in the brain, qPCR results do not support a role for brain aromatase, cyp19a1b, in initiating behavioral sex change, as expression changes for this gene trailed rapid behavioral changes. Brain isotocin expression strongly correlated with TP male-specific behaviors and the rapid behavioral changes characterizing the onset of sex change in the bluehead wrasse, but not the others. Based on these results, authors conclude that while key components of the molecular machinery controlling gonadal sex change are phylogenetically conserved among wrasses, neural pathways governing behavioral sex change may be more variable. I believe that comparison among several species could shed light on the communality and diversity in sex change mechanism. So, I recommend that this manuscript would be acceptable after solving my following concerns.


In this study, I think that non-breeding ovary in temperate species is one of the important points. Authors also describe that there is ambiguity between these gonads (L. 481-482). Probably, there could be atretic oocytes in not only ET gonad but also non-breeding ovary. So, what criteria do authors apply to distinguish between non-breeding ovary and ET gonad? I recommend that histological picture showing the criteria which readers can distinguish should be added in figure 5. Ambiguous classification leads to misinterpretation of the present results. Especially, in discussion section of cyp19a1a, authors discussed that downregulation of its expression is an important event in the initiating stages of gonadal sex change in temperate species also. But, it has been reported that there is no difference of estrogen level between sexes at non-breeding season in protogynous species exhibiting seasonal breeder (Bhandari et al., 2003; Ohta et al., 2008). These facts indicate that cyp19a1a expression would be enough low level in non-breeding female. In the present study also, Fig. 7C and 7D does not show significant difference between each stage. Thus, I do not fully agree that cyp19a1a were downregulated at the initiation of gonadal sex change in temperate species.


Authors emphasized that amh act as an important initiator of male phenotype during female-to-male sex change, while it seems to me that amh expression upregulated after appearance of testicular tissue (specifically appearance of spermatid or spermatozoa). Similar results were reported in a protogynous fish (Horiguchi et al., 2018). Horiguchi et al., 2018 discussed that amh plays a role in spermatogenesis rather than initiator of male phenotype during protogynous sex change. I think that change in amh expression should occur before testicular appearance (St 3 at the latest) if amh plays an important role in initiator of testicular differentiation. What do authors think about this?


Specific coments
L. 235. There is no “Untergasser et al., 2012” in references list.

L. 277. In Table S5, there is no description about the choice of reference genes.

L. 280. I could not see anything about the explanation of normalization in Table S5.

L. 345. Authors explain that figure 5 exhibits gonad histology in Survey 1 (non-breeding season sampling). But, I wonder why Fig. 5 includes ovary of breeding season, not non-breeding. I recommend that histological picture of non-breeding ovary should be added in Fig. 5.

L. 379. Number of degrees of freedom is “4”, isn’t it?

L. 504-507. This sentence is no needed, I think.

Reference No 21 and 22 were not cited in the manuscript.

I think that most figures meet the standard quality of images. However, histological picture in Fig 6A includes something contamination and degree of staining in Fig 6B is absolutely different. I recommend that these histological pictures should be replaced.

·

Basic reporting

The manuscript Conservation and diversity in expression of candidate genes regulating socially-induced female-male sex change in wrasses by Thomas and colleagues presents a comparative study of the mechanisms underlying sex change in 3 species of protandrous fishes. The study focuses on 4 logically selected candidate genes and presents both positive and negative results. This is a useful first study of this group of fish. The paper is very clearly written, well organized, and effectively packaged with appropriate citations and discussion. The authors MUST check the text to be sure the right supplemental table is indicated at each point (see comments 1-3 but also check the rest of the document).

1) Line 265 direct the reader to Table S4 for the annealing temperatures but that information is actually in table S3.

2) Line 277 directs the reader to Table S5 for information about chosen reference genes but that information is actually in table S6.

3) Line 280 directs the reader to Table S5 for information about the differences between normalization strategies, but table S5 gives primer sequence, amplicon size and primer efficiency.

4) Line 248-249 directs the reader “below” for details regarding the use of heterologous oligos but I find additional information only in the supplement, not below (or I’ve missed it).

Experimental design

The authors provide all raw data and R code used to analyze that data. They also provide an incredibly thorough MIQE table in the supplement.

5) I cannot find a supplemental table that demonstrates the different influence of normalization strategies. Comparing the main figures to the supplement figures, I don’t see a drastic change in the trend of the results. I am confused by lines 275-280. Perhaps by presenting the results in supplement with additional explanation this will be clear?

Validity of the findings

6) I find it interesting that the isotocin expression of the IP males was similar to that of the females. This results supports its hypothesized role in establishing dominance. The result is stated on line 439, but I would think it could be emphasized in the discussion.

7) I would encourage the authors to make the sentence about isotocin decrease in gobies (line 538-540) explicitly state that it was a decrease in cell number. This is a minor detail but because some reports relate to cell size and some to cell number it is nice to have this information provided explicitly.

Additional comments

Thank you for an enjoyable read.

---

## Round 0.2 · accepted · Accept

Dear Dr. Thomas and colleagues:

Thanks for revising your manuscript based on the minor concerns raised by the reviewers. I now believe that your manuscript is suitable for publication. Congratulations! I look forward to seeing this work in print, and I anticipate it being an important resource for research communities studying factors governing female-male sex change in wrasses, as well as reproductive biology in general. Thanks again for choosing PeerJ to publish such important work.

Best,

-joe

# ·

Basic reporting

NA

Experimental design

NA

Validity of the findings

NA

Additional comments

The majority of my initial comments were technical issues with table numbers etc. and those have all been resolved.
The authors have also appropriately added the details that I requested concerning isotocin expression in other studies.
I am excited to see this paper in press.
I made a couple notes in the resubmitted version that might be attended to prior to copy editing stages.
Line 74 mentions the results of aromatase inhibitors in "teleosts" but the papers cited are only sex changing teleosts.... If this result is generalizable to other fish, there should be reference to them, or clarify this sentence to be more specific.
Line 82 The word "become" is vague, do you mean independently co-opted in evolution or do you mean "has been identified as"
Lines 83 should "dmrt1" be "dmy" here as it refers specifically to the fish homolog? or possibly indicate "teleost homolog of dmrt1"
Line 502 the species name should be italicized.
Line 529 Cyp10a1b is included in a list that is preceded by the descriptor "neuropeptide" but it is not a neuropeptide.... just reword this sentence.
(I apologize for the delay in my review)